# Impact of Becoming a Certified Oncologic Center of Pancreatic Surgery: Evaluation of Single-Center Perioperative Results and Quality of Life Before and After Implementation of a Certified Center

**DOI:** 10.3390/diseases13110353

**Published:** 2025-10-31

**Authors:** Jan-Paul Gundlach, Thorben Fedders, Steffen Markus Heckl, Thomas Becker, Julius Pochhammer

**Affiliations:** 1Department of General, Visceral-, Thoracic-, Transplantation- and Pediatric Surgery, University Medical Center Schleswig-Holstein (UKSH), Campus Kiel, Arnold-Heller-Strasse 3 (Building C), 24105 Kiel, Germany; 2Department of Internal Medicine II, University Medical Center Schleswig-Holstein (UKSH), Campus Kiel, 24105 Kiel, Germany

**Keywords:** pancreatic ductal adenocarcinoma, PAC, PDAC, pancreas resection, pancreatic cancer center, German Cancer Society, quality of Life

## Abstract

Background: Centralization and certification mark constant processes in everyday clinical routine. Despite the continuously rising number of certified pancreatic cancer (PAC) centers in recent years, fewer than 40% of PAC resections are still performed in certified institutions nationwide. The main objective of the certification is the enhancement of patient survival. Furthermore, certification is intended to improve structural quality, multidisciplinary cooperation, and the transparency of treatment pathways. In addition, it should have a positive effect on patient satisfaction. However, it requires the substantial effort of all partners involved. We aim to illustrate both advantages and limitations of the certification process. Methods: We analyzed perioperative outcomes of patients undergoing pancreatic resection for PAC (ICD C25) before and after our center’s first certification, and benchmarked these results against national data from the German Cancer Society. In addition, we analyzed quality of life (QoL) longitudinally using the validated QLQ-C30 questionnaire administered preoperatively and at 1, 4, and 18 months postoperatively. Results: The study cohort included 47 patients treated in the three years prior to certification and 130 patients during the subsequent seven years as a certified center. The mean annual number of PAC resections increased from 15 (ranged 14–18) to 19 (ranged 10–26). In-hospital mortality, length of stay, and rate of exploration-only procedures remained unchanged. Indicators of procedural quality, such as the number of harvested lymph nodes (*p* = 0.1485) and the precision of histopathological assessment, improved slightly but not significantly. QoL scores generally improved after discharge in both groups; however, functional scales and symptom measures demonstrated unexpectedly inferior values following certification, possibly reflecting higher case complexity. Conclusion: Achieving and maintaining certification requires substantial and continuous effort from all disciplines involved. While major improvements in morbidity, mortality, and long-term QoL were not observed, certification ensured clearer delegation of responsibilities, standardized documentation, and structured quality control. We therefore consider the certification process valuable for promoting multidisciplinary collaboration, maintaining high treatment volumes, and ensuring transparent oncological care pathways.

## 1. Introduction

To date, radical tumor resection is regarded as the only viable option for long-term survival or even a cure for pancreatic cancer (PAC). The surgical quality of pancreatic resection has been addressed by numerous studies [1,2]. Perioperative mortality has significantly decreased in the last 30 years; nevertheless, morbidity remains high at 30–50%, irrespective of high-volume centers [3,4]. Continuous improvement in surgical technique, perioperative intensive care, complication management, and a documented correlation between high case load and a low perioperative morbidity and mortality have caused a trend of centralizing pancreatic surgery [2,5]. Certified centers of excellence have set a new standard for the quality and safety of therapy. Centers of excellence are considered to have high-procedure volumes and dedicated specialists, both of which are certified by independent quality control. Certified patient care is standard in everyday practice, but the clinic to be certified is faced with great burdens. In 2003, a certification system was established by different expert associations of the German Cancer Society (Deutsche Krebsgesellschaft—DKG) [6]. In the subsequent period, different tumor entities were added. By 2012, 42 centers had been certified for PAC—with a rising trend: the latest annual analysis by the German Cancer Society revealed 166 certified PAC sites at the end of 2024 [7].

### 1.1. Surgical Case Numbers of German Pancreatic Cancer Centers

For the recent certification period of 2024, 7.239 newly diagnosed primary cases were treated for PAC in a certified cancer center with a median of 45 cases per center [7]. Thereof, 3.078 patients (42.5%) were resected. Patients were classified in Union internationale contre le cancer (UICC) stages I–IV. In total, 277 cases were included in IA, of which 77.3% were operated. The biggest group of surgically treated patients is, as in previous years, reflected by UICC IIA and IIB. However, the largest group of newly diagnosed PAC cases is represented by group UICC IV with 3.333 cases, of which only 6.3% were operated on.

In 2023, a median of 19.5 patients were operated on (OPS 5-524ff and 5-525ff) for PAC (ICD1-10 C25) per certified site with a range of five to seventy-seven cases. In total, 7.184 pancreatic operations were performed irrespective of their indication in certified centers in 2023. Interestingly, in 2018, while a total of 5.683 patients with PAC were treated in a cancer center in Germany, this marks only 30.9% of all cases in Germany with respect to the incidence of C25 PAC in Germany in 2016 [8]. Based on the current status of patients treated in a certified center for the key figure year of 2023 and on the assumption of unchanged incidence of PAC, it can be assumed that still more than 50% of patients are treated in non-certified centers in Germany nowadays.

### 1.2. Surgical Experience for Pancreatic Cancer Centers

Certification policy requires clinical standardization. The relationship between resection volume per hospital and its respective postoperative mortality is unmistakable [9,10] and represents an adequate tool for quality control. Thus, becoming a certified PAC center requires certain numbers of surgical cases per year. Expertise is demanded by the German Cancer Society for pancreatic surgery with at least 20 pancreatic resections per year, of which 12 cases represent primary PAC (adenocarcinoma and neuroendocrine carcinoma) cases. Two surgeons are required per site with a respective minimum requirement of 10 resections per year. In addition, integrated patient care is required with multidisciplinary hospital treatment (implemented with a tumor conference once per week) as well as collaboration with general practitioners, psycho-oncology, physiotherapy, patient welfare, pastoral work, nutrition counseling, and oncology outpatient service. In addition, profound tumor documentation is required.

We hereby give insights into our surgical experiences of becoming a PAC center and contrast our expertise before and thereafter by analyzing patient data over a period of three and seven years before and after becoming a certified center, respectively, and contrast certification struggles. Furthermore, we analyze the expected benefits for patients. Since quality of life (QoL) examinations represent a useful addition to therapy evaluation in pancreatic surgery due to the severity of the procedures with a relevant risk of complications [11], we added an analysis of global health status, diverse functioning scales, and different symptoms, as well as items for financial difficulties caused by the cancer. This appears particularly relevant in view of the limited life expectancy of patients with PAC [2,12].

## 2. Materials and Methods

In 2012, our university hospital became a certified center for PAC by the German Cancer Society. We retrospectively identified all discharges for pancreatic surgery using appropriate procedure codes for pancreas resection (5-524 et seq.) and for total pancreatectomy (5-525 et seq.) from the OPS as published by the German Institute for Medical Documentation and Information (DIMDI). To solely identify resections for malignancy, we restricted our sample to patients who had an accompanying PAC diagnosis using the International Classification of Diseases Code ICD-10 C25. We analyzed these patients and divided the cohort into those treated before and after the center achieved certification. All patients included in this study signed informed consent and samples were used in accordance with institutional guidelines and the Declaration of Helsinki. Resected tumors were classified according to the TNM staging system of the International Union Against Cancer.

For QoL assessment, patients received a questionnaire based on the European Organization for Research and Treatment of Cancer (EORTC) QLQ-C30 questionnaire prior to operation and at 1 month, 4 months, and 18 months afterwards. The EORTC QLQ-C30 is a cancer-specific questionnaire consisting of 30 items, which cover self-perception scales of functioning (physical, role, cognitive, emotional, and social), a global health scale, eight different symptoms (fatigue, nausea/vomiting, pain, dyspnea, insomnia, loss of appetite, constipation, and diarrhea), as well as items for financial difficulties caused by the cancer. All items are scored in response categories with four levels, except for two global health items which are scaled with seven levels from very poor to excellent. High scores for the functional and global health scores represent high response and high scores in the symptom categories, which represent a high level of dysfunction. Our results were compared to a population-based normative sample from the Northern German population [13].

### Statistical Analysis

Statistics are represented using mean ± standard deviation (SD). Overall survival was analyzed by the Kaplan–Meier analysis and statistical evaluations were performed by log-rank test. The data was tested for normal distribution and equal variances using Shapiro–Wilk. For normally distributed data, significances were calculated using T-test. For multivariate survival probability testing, Cox regression was used and *X*^2^ test was used for contingency. HRQoL data were pooled for 4 months (3 and 6 months) and 18 months (12 and 24 months) postoperatively and analyzed according to the EORTC-scoring manual [14]. Missing QoL data were handled according to the manual, with partial questionnaires retained if ≥50% of items per scale were completed; completely missing datasets were excluded. Statistical significance was defined at a *p*-value of <0.05. Statistical analysis was conducted using SPSS software version 29 (IBM, Armonk, NY, USA). Graph Pad Prism 9.5.0. (San Diego, CA, USA) was used to present data.

## 3. Results

We analyzed a cohort of 47 (21 female, 44.68%) and 130 (67 female, 51.15%) patients who received pancreatic resection for oncologic indication during the time before becoming a certified center and thereafter. Table 1 gives detailed information on patient characteristics, histopathological findings, UICC scoring, and the status of resection (R0: no microscopic residual tumor; R1: microscopic residual tumor; R2: macroscopic residual tumor) for the cohort before becoming a certified center (A) and thereafter (B).

Furthermore, we indicate the duration of hospitalization and state death within 30 days from operation. Although the mean number of operations per year increased from 15 (14–18) to 19 (10–26) in the observational period, we found no significance in this deviation (*p* = 0.1859). Mean age of operated patients did not significantly differ (64.6 ± 11.0 vs. 63.7 ± 15.2; *p* = 0.4214). Patients were hospitalized for a median of 19 days (ranged 8–74 days) before and 19 days (ranged 0–68) after becoming a pancreatic center, which did not differ significantly in the two groups, respectively (*p* = 0.3346). Furthermore, overall survival did not differ significantly in the two groups either (*p* = 0.3638; Figure 1A). Age at operation (OR = 1.02, [0.98; 1.06], *p*= 0.346) or duration of hospitalization (OR = 1.02, [1.0; 1.05], *p*= 0.0565) did not influence overall survival; Figure 1B,C demonstrate UICC distribution in the two cohorts, which did not differ significantly between the groups (*x*^2^ = 1.574, *p* = 0.8134). The proportion of patients receiving neoadjuvant treatment was low in both cohorts (4% vs. 2%). Furthermore, laparoscopic procedures in our cohort were limited to distal pancreatectomies.

In 25.40% and 24.14% (mean 5.33 vs. 6.00 cases/year), exploration only without resection were performed before and after becoming a certified center due to unresectable vascular infiltration or metastatic spread (Table 2).

### 3.1. Comparison of Surgical Performance Characteristics

The number of resected lymph nodes improved over the years, although not significantly: before becoming a certified center, in 10 out of 39 adenocarcinoma cases, we found less than 10 resected lymph nodes (25.64%). In the second period, 15 cases with an insufficient number of resected lymph nodes were found (14.29%; less than 10 lymph nodes were resected in 14 cases from 2012 until 2016 and one aberrant case was found thereafter, when TNM classification was modified to 12 required lymph nodes for acceptance of N0 status in the eighth edition in January 2017). The mean number did not differ significantly (16.47 vs. 18.32, *p* = 0.1485; Figure 1D). While tumor-free resections (R0s) were achieved in 78.72% (37 cases) and 78.46% (102 cases) of the cases for the time before and after becoming a PAC center, R1 resection increased from 8.51% to 19.23%.

### 3.2. Analysis of Perioperative Morbidity and Mortality

Superficial surgery site infections (category A1 as defined by the Robert Koch society) were found in 23.40% (11/47) and 9.76% (12/130) of the cases within 30 days from operation (Figure 1E). Revision surgery was performed in 27.66% and 17.07% of the cases within the respective hospital stay (range up to 74 days). The in-hospital mortality was 4.26% and 4.62% for the two groups.

### 3.3. Quality of Life Assessment

QoL assessments reveal in both groups increased global health status (Figure 2A) after perioperative hospitalization; however, it remained below the population-based normative sample. In addition, functional QoL scales (Figure 2B–F) also increased after an initial decrease during the early postoperative time (discharge) in both groups. Interestingly, a trend supporting superior QoL status before becoming a certified PAC center is visualized, although only in physical and emotional functioning, whereby a significant difference (*p* = 0.0368 and *p* = 0.0142) is demonstrated 18 months after operation (Figure 2B,E). In general, 18 months after the operation, functioning has improved in almost all parameters. Although the improvement was greater in the group of patients before becoming a certified center than in the following group, global health status and functioning scales remained throughout the study period below the normative sample, except for global health status and emotional functioning 18 months after operation, which were clearly above the normative sample. Although not significantly, financial difficulties were higher for patients after surgery in the recent period (Figure 2G).

Next, we analyzed special symptoms tested by the QoL questionnaire (Figure 3). Of note, patients suffered from fatigue significantly more often in the second group in a long-term perspective (*p* = 0.0076). Nausea/vomiting and appetite loss before operation were significantly less experienced in the first group (*p* = 0.0136 and *p* = 0.0067, respectively, Figure 3B,F). In general, symptoms were highest at discharge and decreased thereafter, except pain (Figure 3A), insomnia, and fatigue (Figure 3C,D) in the post-certification group, which remained at the discharge level.

## 4. Discussion

Although the importance of high-volume centers and surgical experience for postoperative outcome are recognized [9,10], centralization is proposed to be based not only on dependence on surgical volume but also on multidisciplinarity in the hospital [15]. The certification program by the German Cancer Society has led to several paradigm shifts in Germany. Most importantly, multidisciplinarity was widely established in patient treatment, expanding beyond medical discipline. Furthermore, standardized documentation allows benchmarking [6]. Since we did not find significant improvements in morbidity, mortality, and QoL in our study, we highlight potential benefits of certification in terms of structured care pathways, standardized processes, and improved documentation, which may influence specific outcomes and perioperative management. In the following, we benchmark our data with German Cancer Society data, which is intended only to provide context, without implying statistical equivalence.

### 4.1. Apart from Missing Surgical Changes, Patient Treatment Improved Significantly

Clinical improvements and mid-term benefits in surgical PAC care should, however, be set against the strenuousness of the certification process in both the short and long run of becoming and remaining a certified center. In detail, additional personnel for administration were appointed (two assistants) at our department. Certification processes are often accompanied by the implementation of standardized care pathways, structured monitoring, interdisciplinary routines, and intensified staff training—all factors known to reduce complication rates and improve symptom control. At our institute, implementation of multi head conferences (tumor boards) with all involved departments was significantly raised in the first years after certification; hence, a profound assessment of the operability was realized and a stringent selection of patients resulted. Furthermore, implementation of morbidity and mortality (M&M) conferences as feedback instruments through peer review will in all probability prevent repetition of errors leading to complications. The surgical procedure has been harmonized, with only two surgeons primarily performing the interventions. Furthermore, laparoscopic procedures in our cohort were limited to distal pancreatectomies. No minimally invasive pancreatic head resections were performed during the observation period, as these were not part of the standardized surgical portfolio at that time. Interestingly, as stated by their respective directors, stomach cancer centers improved in quality after first certification by 87.9%. However, in light of repeated certifications, it is stated that the expenses exceed the benefits and no financial advantages occur [16], most likely because of unstable refinancing and considerable administrative input. At our department, certification did not magnify attractability, as the number of operations and DRGs did not significantly increase. Nevertheless, the certification process ensures a clear delegation of responsibilities and competences, and a constant high level of quality; it also reduces coordination needs. In addition, improved perioperative settings in pancreatic surgery with better outcome will consequently lead to reduced costs [17]. In general, although surgical quality did not improve considerably, treatment quality was improved on multifaceted aspects. Certification will lead to centralization and higher case volumes per surgeon and team, which have been repeatedly associated with improved perioperative outcomes in pancreatic surgery [2].

### 4.2. Comparison of Surgical Performance Characteristics

For PAC surgery, different indicators of quality exist [18,19]. A beneficial marker for procedural quality is the length of hospital stay. This assumption is supported by the fact that complications after surgery will most probably cause a prolongation of the hospitalization. In Austria, as well as in Germany, the median length of stay was reported to be 16 days [20,21], while we found a median length of stay of 19 days in both our cohorts, respectively. An objective quality feature of oncologic surgery is considered by the number of resected lymph nodes [22], which was significantly improved in our cohort over the years. While 25.64% of the cases were found with insufficient lymph nodes resected before becoming a certified center, this decreased to an average of 14.29% thereafter and is currently further decreasing. Meanwhile, the essential number of resected lymph nodes is still under debate [23]. Of note, high rates of R1 resection in PAC are not a marker of low-quality surgery but rather of high-quality pathology [24]. The latest German guideline from 2024 recommends a margin of >1 mm (i.e., CRM negative, R0 wide). However, due to a more dispersed tumor growth compared to other tumor entities such as rectal cancer, this margin is postulated to be too little to represent R0 [25] and a 1.5 mm margin is proposed for R0 acceptance [26]. Still, there is no consensus on the pathological assessment of microscopic margin involvement and standardized techniques [24]. For the German Cancer Society, 79.0% of cases (median of 15 cases with a range of 5–62 cases) with R0-resection were reported [7]. Thus, our R0 results (tumor-free resection in 78.72% (37 cases) and 78.46% (102 cases) of the cases for the time before and after becoming a PAC center, respectively) are similar to those obtained by the German Cancer Society. The proportion of R1 resection as a marker of pathological examination quality increased from 8.51% to 19.23%, whereas undefined Rx status was extinguished from histopathological reports due to standardized processes, which could be a potential advantage, as compared to definitive improvements in clinical endpoints.

### 4.3. Analysis of Perioperative Morbidity and Mortality as Well as Quality of Life Assessment

In our cohort, morbidity remained stable, as comparable to the literature [3,4]. However, it must be considered that, especially in the first cohort, the data situation limited a detailed evaluation of, for example, the administration of blood transfusions or postoperative drug treatment of surgery site infections and pancreatic fistulas. For the German Cancer Society group, surgery site infections were reported in 3.9% of the entire (oncologic and non-oncologic) resection group [7]. We found surgery site infections in the two groups in 23.40% (11/47) and 9.76% (12/130) of the cases (category A1: superficial wound infections) within 30 days from operation, respectively. Of note, patients with pancreatic carcinoma have a higher incidence of postoperative infectious complications than those with intraductal papillary mucinous neoplasm or cystic tumors [27]. Revision surgery and mortality are accepted markers of quality [19]. Revision surgery was performed on 27.66% and 17.07% of the cases within the hospital stay (ranged up to 74 days) at our clinic, compared to 9.4% of the cases within 30 days for the German Cancer Society [7]. While reasons for revision were stated as anastomosis insufficiency, secondary bleeding, and pancreatitis [7], our revisions also include surgery site infections and burst abdomen. Mortality rates were comparable to that obtained by the German Cancer Society. Our overall survival analysis for patients after pancreas resection showed no relevant differences between the two groups, respectively. The in-hospital mortality was at 4.26% before becoming a certified center. After certification, 30-day mortality was at 4.62%, which is nearly comparable with the German Cancer Society (3.6%) [7]. The proportion of patients receiving neoadjuvant treatment was low in both cohorts (4% vs. 2%), reflecting the still limited role of neoadjuvant approaches during the study period. At that time, neoadjuvant therapy was not routinely established for borderline resectable PAC at our center. Current treatment strategies, however, increasingly emphasize neoadjuvant regimens for borderline and locally advanced disease, which will likely affect future perioperative and oncological outcomes.

QoL assessment was performed to analyze possible differences in physical and mental health as well as in global health status. In the literature, pancreatic surgery itself is reported to contribute to a reduced QoL [28]; nevertheless, several reports exist on early recovery to baseline levels 6 months after operation [12,29,30]. Interestingly, even perioperative complications were reported to have little influence on QoL recovery [31]. In addition, the resection method showed little influence: classical Whipple vs. pylorus-preserving Whipple procedures [32,33,34] as well as total pancreatectomy did not show differences in QoL [35], although this finding is fluctuating [28]. In our cohort, after initial postoperative increment, global health status and functional QoL scales increased until 18 months after operation. Of note, physical and emotional functioning revealed significantly higher improved recovery in the group before becoming a certified center than thereafter. In addition, financial difficulties were higher for patients after surgery in the recent period. In general, it can be seen that most of the different functioning scales are marginally lower in the group after first certification. Documentation effects and shifts in case mix (e.g., selective referral of more complex cases or more detailed assessment of postoperative limitations) may influence the measurement of functional scales. Particularly, documentation effects may explain why functioning scores appeared lower in the certified cohort. Moreover, modern treatment regimen causes higher rates of treatment-related toxicities. Before operation, nausea/vomiting and appetite loss were significantly higher in the group after certification (*p* = 0.0136 and *p* = 0.0067). Although there was a tendency for symptoms to persist 18 months after discharge in the second cohort, this can only be demonstrated significantly in the case of fatigue (*p* = 0.0076). In our cohort, QoL measures might have been influenced by whether patients received adjuvant chemotherapy and by the choice of regimen. Until 2018, adjuvant treatment of PAC adenocarcinomas consisted mainly of single-agent chemotherapy with gemcitabine [36] or 5-fluorouracil [37], with gemcitabine often favored due to its comparatively favorable tolerability profile. The publication of the PRODIGE-24/CCTG PA.6 Phase III trial in 2018 represented a paradigm shift, demonstrating a significant survival benefit of adjuvant-modified FOLFIRINOX (5-fluorouracil, leucovorin, irinotecan, oxaliplatin) over gemcitabine in patients with resected pancreatic cancer [38]. However, this benefit comes at the cost of a substantially higher rate of treatment-related toxicities: symptoms such as fatigue, diarrhea, nausea and vomiting, abdominal pain, and sensory neuropathy were significantly increased in the modified FOLFIRINOX group in comparison to the gemcitabine monotherapy group. Both regimens extend over six months, but it is important to note that chemotherapy-associated fatigue and sleep disturbances in particular may persist well beyond treatment completion, and in some patients remain a long-term burden [39], which might explain the QoL results for postoperative symptoms in the cohort after becoming a certified center.

### 4.4. Certified Centers Have a Pioneer Role and Other Hospitals Will Follow

As indicated above, less than 40% of PAC resections are performed in a certified center nationwide [7,40]. Nevertheless, PAC centers have a leading role, and uncertified centers will most probably adopt procedural improvements; thus, the quality between certified centers and other departments will be equalized for the good of the patient`s outcome. In general, reasons for better outcomes at cancer centers compared to smaller institutes might be diverse. Patients at cancer centers are more likely to be considered in multidisciplinary conferences and they might undergo more sophisticated preoperative imaging tests. In addition, surgeons at cancer centers may be more specialized compared to non-cancer center hospitals [41]. Nevertheless, another rationale of cancer centers is to unite high-volume centers of pancreatic surgery in order to recruit patients for interdisciplinary trials, such as neoadjuvant regimens, where randomized controlled trials are lacking so far [42]. Standardized neoadjuvant regimens would be more likely to raise the total number of pancreatic resections [43,44], improve tumor-free margins (down-staging) [44], and possibly reduce postoperative complications [45,46]. It is important to note that, regardless of centralized care, there is an urgent need to detect the disease early. Modern miRNA techniques can provide a possible route out of the dark [47]. Apart from certification by the German Cancer Register, the German Society for General and Visceral Surgery (DGAV) has launched a registry for quality control, risk assessment, and outcomes research in pancreatic surgery in Germany (DGAV StuDoQ|Pancreas) [40]. Participating centers enter data from patients who consented prospectively in a pseudonymized form. Although an increased workload is required for additional registry participation, nationwide studies from the StuDoQ|Pancreas dataset can contribute to improving treatment strategies for PAC [18].

This study has several limitations. First of all, the sample size. In particular, the relatively small pre-certification cohort, which is caused by insufficient data availability before that, results in limited possibilities for applying complex statistical methods. Performing a statistically robust matching procedure would have resulted in a considerable loss of cases. Therefore, data interpretability is limited by the sample size. In particular, our observations of in-hospital mortality and comparable postoperative morbidity should be viewed with caution due to the limited number of cases. Next, the QoL EORTC QLQ-C30 questionnaire mainly reflects general aspects of cancer-related QoL and lacks disease-specific items for PAC. Future studies should include PAC-related aspects. In addition, more elaborate documentation and the possibility that more complicated cases were treated after certification may have influenced certain functional outcomes. We therefore propose additional adjusted analyses to further quantify these mechanisms.

## 5. Conclusions

In conclusion, our clinic, representing a moderately sized PAC program, has considerably improved in patient treatment as a matter of procedural improvement, although no major improvements in surgical experience such as morbidity and mortality as well as QoL were found. Nevertheless, certain aspects such as the increased number of sufficiently resected lymph nodes—although not significant—support improved patient treatment. In addition, the percentage of resection without a residual tumor has been high beforehand, although R1 resection has increased with more sophisticated pathological examinations as well as improved surgical/pathological interaction. Altogether, despite severe laboriousness and presumed uncovered expenses, we positively assess the certification and assume cost effectiveness due to expected better overall outcomes, which needs to be analyzed in future studies with larger cohorts. The participation of surgical departments with complex onco-surgical interventions in clinical multicenter observational studies is a contribution to research on surgical treatment, and it appears reasonable and recommendable since the results of such studies can support decision making processes in daily surgical practice and most likely improve overall survival of this devastating disease.

## Figures and Tables

**Figure 1 diseases-13-00353-f001:**
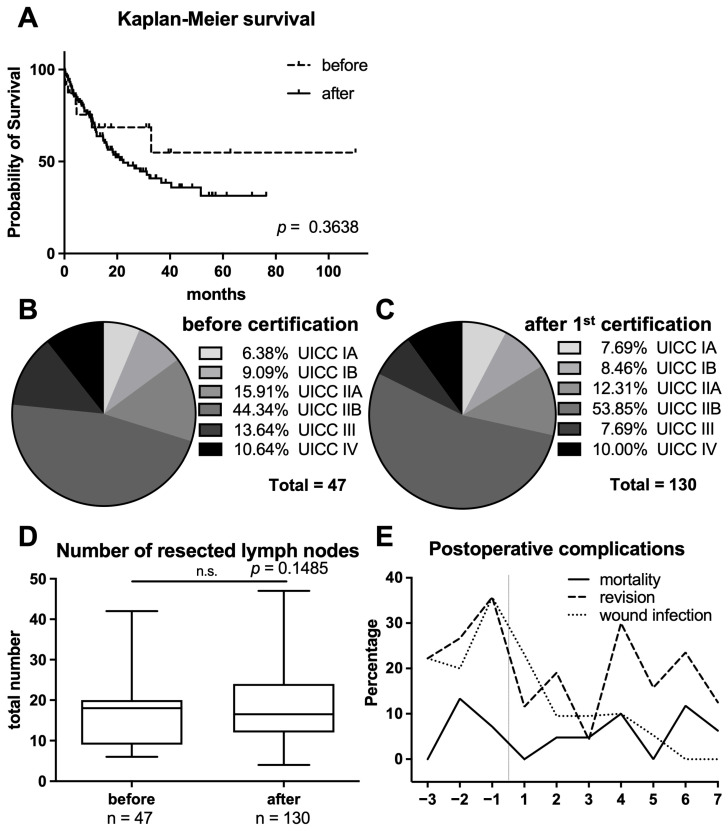
Patient survival, UICC characteristics, number of resected lymph nodes, and postoperative morbidity/mortality. (**A**) Shown are Kaplan–Meier survival curves for patients before becoming a pancreatic cancer (PAC) center (n = 47) and becoming a PAC center (n = 130), without significant differences (*p* = 0.36). (**B**,**C**) Tumor stage classification by the Union internationale contre le cancer (UICC) and distribution within the different stages after histological analyses of resection specimens before (**B**) and after (**C**) becoming a PAC center are displayed, which did not differ significantly (*p* = 0.81). (**D**) Boxplots of the total number of resected lymph nodes for the two groups are demonstrated. The bold strip represents the median of the group. (**E**) The figure provides detailed information on postoperative complications: in detail, the occurrence of mortality, revision surgery, and surgery site infections within 30 days from pancreas resection before and after becoming a PAC center are compared; the interrupted vertical line marks the first certification.

**Figure 2 diseases-13-00353-f002:**
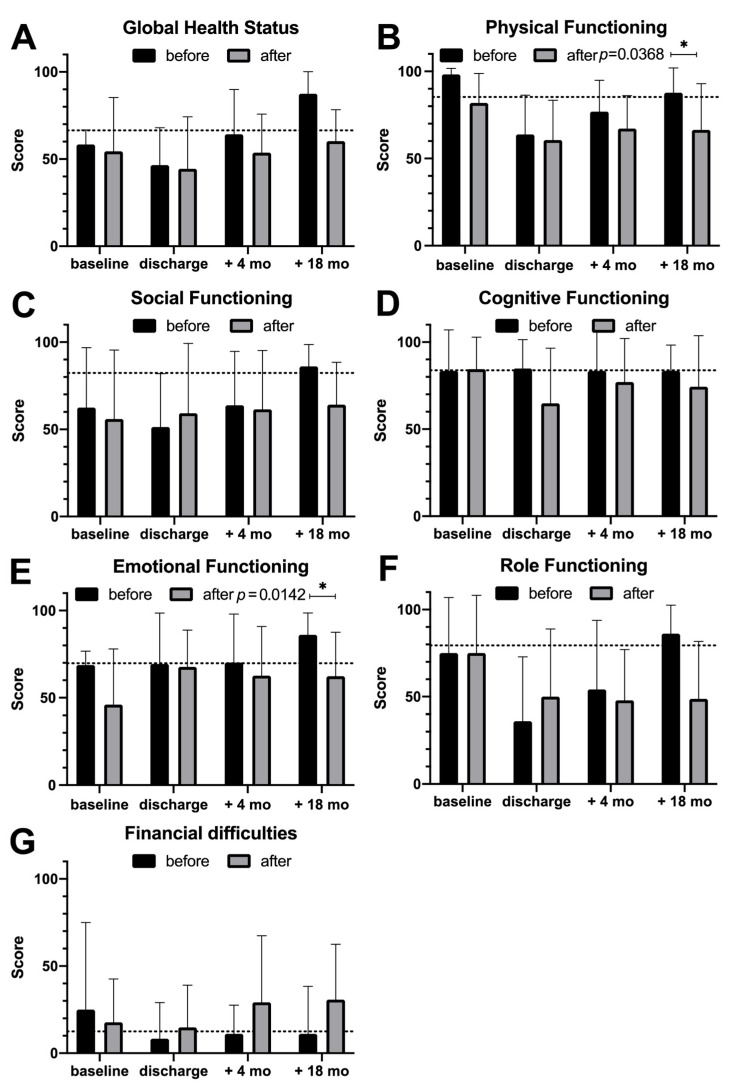
Quality of Life (QoL) assessment and functioning analysis with the QLQ-C30 questionnaire stratified by cohorts. (**A**) Global health status, (**B**) physical functioning, (**C**) social functioning, (**D**) cognitive functioning, (**E**) emotional functioning, and (**F**) role functioning are demonstrated for the time before becoming a certified pancreatic cancer center and thereafter. Visualized are the groups at baseline (before operation), discharge after operation, and after 4 and 18 months. Increased functional QoL scales are visualized after discharge. Eighteen months after operation, physical (**B**) and emotional (**E**) functioning are significantly reduced in the group after becoming a certified center (*p* = 0.0368 and *p* = 0.0142, respectively). Although not statistically significant, financial difficulties were higher for patients after surgery in the recent period (**G**). The dotted line represents a normative dataset. Asterisk indicates statistical significance (*p* < 0.05). Abbr.: mo: months.

**Figure 3 diseases-13-00353-f003:**
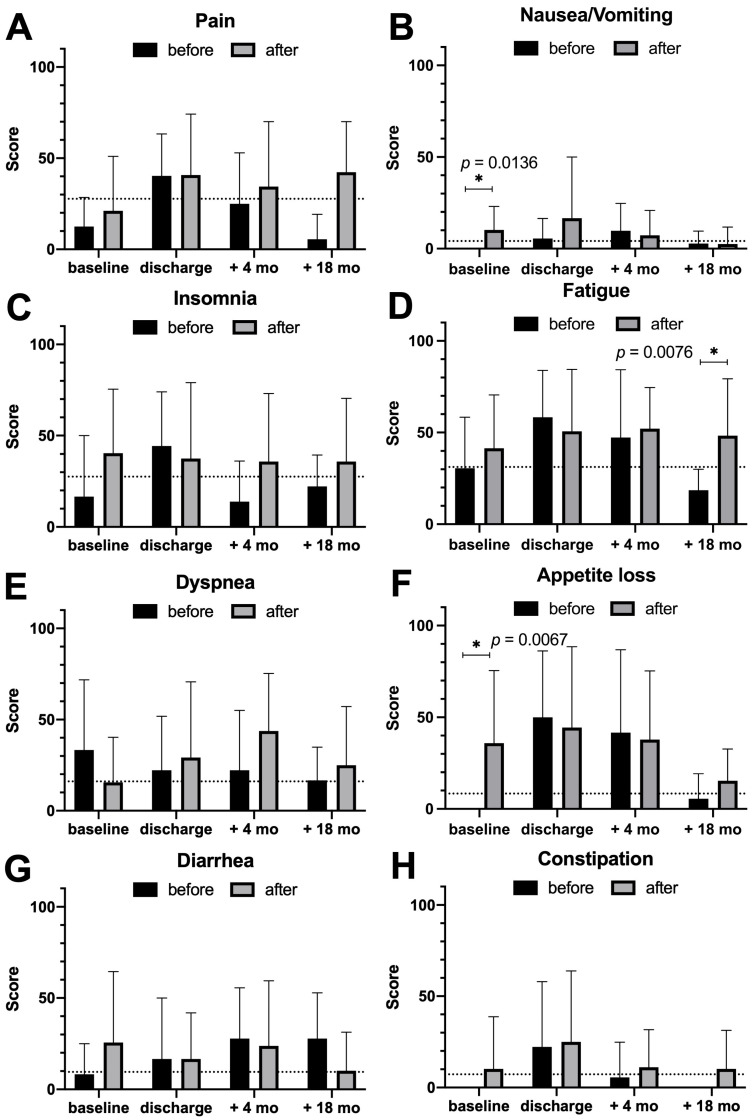
Quality of Life (QoL) assessment of different symptoms with the QLQ-C30 questionnaire stratified by cohorts. (**A**) Pain, (**B**) nausea/vomiting, (**C**) insomnia, (**D**) fatigue, (**E**) dyspnea, (**F**) appetite loss, (**G**) diarrhea, and (**H**) constipation are demonstrated for the time before becoming a certified pancreatic cancer center and thereafter. Visualized are the groups at baseline (before operation), discharge after operation, and after 4 and 18 months. Decreased QoL-analyzed symptoms are visualized after discharge. Before operation, nausea/vomiting and appetite loss were significantly higher in the group after certification (*p* = 0.0136 and *p* = 0.0067). Although there was a tendency for symptoms to persist 18 months after discharge in the second cohort, this can only be demonstrated significantly in the case of fatigue (*p* = 0.0076). The dotted line represents a normative dataset. Asterisk indicates statistical significance (*p* < 0.05). Abbr.: mo: months.

**Table 1 diseases-13-00353-t001:** Patient characteristics, tumor histopathology, UICC stage, resection status, hospital stay, and mortality within 30 days after resection for pancreatic surgery patients before (A) and after certification (B), stratified by year.

**A**				**Histopathology**				**UICC Scoring**						**R-Status**	**Hospitalization**	
year	n	♀	age ± SD	AC	NE	CA	n/a	1A	1B	2A	2B	3	4	0/1/2/x	duration	†
−3	18	8	62.6 ± 9.0	15	3	0	0	1	1	3	7	3	3	15/1/1/1	27.5 ± 21.3	0
−2	15	7	64.5 ± 13.7	13	1	1	0	2	3	2	7	1	0	12/2/1/0	25.9 ± 17.8	1
−1	14	6	67.3 ± 10.9	11	1	1	1	0	0	2	8	2	2	10/1/0/3	28.9 ± 22.3	1
total	47	21	64.6 ± 11.0	39	5	2	1	3	4	7	22	6	5	37/4/2/4	27.4 ± 20.3	2
**B**				**Histopathology**				**UICC Scoring**						**R-Status**	**Hospitalization**	
year	n	♀	age ± SD	AC	NE	CA	n/a	1A	1B	2A	2B	3	4	0/1/2/x	duration	†
1	26	14	66.8 ± 8.3	23	3	0	0	1	3	5	14	1	2	22/3/1/0	27.0 ± 16.8	0
2	21	11	67.7 ± 8.0	16	3	0	2	2	1	3	13	0	2	16/5/0/0	20.9 ± 10.8	1
3	21	11	64.4 ± 12.8	18	3	0	0	1	1	4	12	2	1	15/4/2/0	22.2 ± 11.6	1
4	10	4	70.4 ± 7.0	8	2	0	0	1	3	1	5	0	0	9/1/0/0	29.8 ± 19.3	1
5	19	9	67.8 ± 10.6	11	6	1	1	3	0	1	10	0	5	14/5/0/0	23.9 ± 15.6	0
6	17	11	69.4 ± 10.3	13	2	1	1	2	0	1	10	2	2	15/2/0/0	22.1 ± 9.7	2
7	16	6	59.9 ± 10.5	16	0	0	0	0	3	1	6	5	1	11/5/0/0	23.7 ± 13.2	1
total	130	67	63.7 ± 15.2	105	19	2	4	10	11	16	70	10	13	102/25/3/0	23.3 ± 13.6	6

Values are presented as n or mean ± SD where indicated. Histopathology: AC = adenocarcinoma; NE = neuroendocrine tumor; CA = cystadenocarcinoma; n/a = others. UICC stage: 1A–4. R-status: 0 = R0, 1 = R1, 2 = R2, x = unknown. Hospitalization: mean length of stay in days ± SD; † indicates postoperative mortality within 30 days after operation. While the mean number of PAC operations per year increased from 15 (14–18) to 19 (10–26) in the observational period, we found no significance in this deviation (*p* = 0.1859). Mean age of operated patients did not significantly differ (64.6 ± 11.0 vs. 63.7 ± 15.2; *p* = 0.4214). Patients were hospitalized for a mean of 27.4 ± 20.3 and 23.3. ± 13.6 days postoperatively; however, the median of 19 days (ranged 8–74 days) before and 19 days (ranged 0–68) after becoming a pancreatic center did not differ significantly in the two groups, respectively (*p* = 0.3346). Total values for all years are shown.

**Table 2 diseases-13-00353-t002:** Pancreatic operations for oncologic indication.

	Years Before Certification			Years After 1st Certification						
	−3	−2	−1	1	2	3	4	5	6	7
Total number of patients [n]	25	19	19	29	31	30	15	24	20	23
Completed resection [n (%)]	18 (72.0)	15 (78.9)	14 (73.7)	26 (89.7)	21 (67.7)	21 (70.0)	10 (66.6)	19 (79.2)	17 (85.0)	16 (69.6)
Exploration only [n (%)]	7 (28.0)	4 (21.1)	5 (26.3)	3 (10.3)	10 (32.3)	9 (30.0)	5 (33.3)	5 (20.8)	3 (15.0)	7 (30.4)

Abbr.: n: number.

## Data Availability

The clinical datasets supporting the conclusions of this study were derived from the patient files (paper and electronic form). Therefore, restrictions to availability apply due to data protection regulations. Anonymized data are, however, available from the corresponding author on reasonable request and with permission of the University Hospital Schleswig-Holstein and the local review board.

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
