# Peer review of "Impact of Becoming a Certified Oncologic Center of Pancreatic Surgery: Evaluation of Single-Center Perioperative Results and Quality of Life Before and After Implementation of a Certified Center"

_diseases, 2025, doi:10.3390/diseases13110353_

Round 1
Reviewer 1 Report
Comments and Suggestions for Authors
The core content of this paper is to explore the impact of becoming a certified pancreatic cancer (PAC) surgical center on the perioperative outcomes and quality of life of patients. The study evaluated the advantages and limitations of the certification process by comparing the situations before and after the center obtained the certification. My comments are as follows:
- The sample size was relatively small. Before certification, there were only 47 patients, and after certification, there were 130 patients. Moreover, the longest follow-up period was 18 months.
- There may be selection bias in the patient population before and after certification. The author should detail the differences in baseline characteristics between the two groups of patients and use methods such as propensity score matching to balance the baseline characteristics of the two groups of patients, in order to reduce the influence of potential confounding factors on the results.
- Although the QLQ-C30 questionnaire was used, this questionnaire mainly focuses on the general quality of life of cancer patients and may lack specificity for patients undergoing pancreatic cancer surgery.
- When comparing the surgical quality indicators, complication rates, etc. before and after certification, only simple statistical methods such as t-tests were used. It is suggested that the author adopt more complex statistical models, such as multivariate regression analysis, to control potential confounding factors and more accurately assess the impact of certification on the results.
- The article does not mention how to handle missing data. It should detail the methods for handling missing data and assess the potential impact of missing data on the results.
- Although the changes in quality of life scores are statistically significant, there is a lack of explanations for the clinical significance of these changes. It is suggested that the actual impact of these score changes on the quality of life of patients be discussed in light of the actual clinical situation.
- The analysis of the results in the discussion section is rather superficial. It is necessary to delve deeply into the causes and mechanisms of changes in surgical quality, complication rate and quality of life before and after certification. Combined with relevant domestic and foreign research, the significance of these results should be further elaborated.
- The conclusion section holds that the certification process is valuable for improving the quality of treatment, but the basis for this conclusion seems insufficient.
- The introduction part provides a relatively brief overview of the background knowledge. It is necessary to further supplement information such as the epidemiological data of pancreatic cancer, the current situation and challenges of surgical treatment, etc., to enhance the background depth of the article.
Author Response
Reply to referee #1
We thank the referee for the constructive review. Here, we address all topics point by point.
Comment 1: The sample size was relatively small. Before certification, there were only 47 patients, and after certification, there were 130 patients. Moreover, the longest follow-up period was 18 months.
Response 1: Dear reviewer, indeed, the longest follow-up period was 18 months, which we agree is relatively short. However, the primary objective of our study was to evaluate perioperative outcomes and early postoperative quality of life before and after certification rather than long-term QoL. Giving nature of the disease and corresponding with our Kaplan-Meier survival analysis, after 18 months around 50 % of the patients died. Therefore, the chosen follow-up period was considered sufficient to capture relevant short- and mid-term effects of center certification on patient recovery and quality of life.
Regarding the sample size, we acknowledge that the number of patients before certification (n = 47) was limited compared to the post-certification group (n = 130). This difference is caused by a change of the hospital information system and therefore limited availability of earlier data. Despite this, the results consistently demonstrated meaningful trends, supporting the robustness of our conclusions.
Comment 2: There may be selection bias in the patient population before and after certification. The author should detail the differences in baseline characteristics between the two groups of patients and use methods such as propensity score matching to balance the baseline characteristics of the two groups of patients, in order to reduce the influence of potential confounding factors on the results.
Response 2: We agree with the reviewer, that a propensity score matching would be beneficial. But since these are real world data, we are limited in application of statistical methods: Propensity score matching could not be performed for several reasons. First, due to the limited sample size, especially in the pre-certification cohort (n = 47), performing a statistically robust matching procedure would have resulted in a considerable loss of cases and therefore a significant reduction in statistical power. Second, the data completeness before certification was limited owing to the change of the hospital information system, leading to incomplete documentation of certain baseline parameters required for matching. Third, the main purpose of this study was to describe real-world changes in perioperative outcomes associated with certification, rather than to establish causality. Thus, we aimed to provide an unadjusted yet transparent comparison reflecting the actual patient populations treated before and after certification. We added therefore a paragraph in the limitation section: “This study has several limitations. First of all, the sample size. Especially the pre-certification cohort is relatively small which is caused by insufficient data availability before that, resulting in limited possibilities for applying complex statistical methods. Performing a statistically robust matching procedure would have resulted in a considerable loss of cases.”
Comment 3: Although the QLQ-C30 questionnaire was used, this questionnaire mainly focuses on the general quality of life of cancer patients and may lack specificity for patients undergoing pancreatic cancer surgery.
Response 3: Thank you for pointing this out. The EORTC QLQ-C30 is widely used and mainly reflects general aspects of cancer-related quality of life and lacks disease-specific items for pancreatic cancer. We implemented this questionnaire because it is well-validated and widely used in the literature as well as an accepted instrument for assessing overall quality of life for all oncologic patients. We acknowledge this limitation and have added a corresponding statement to the discussion section: “Next, the QoL EORTC QLQ-C30 questionnaire mainly reflects general aspects of cancer-related QoL and lacks disease-specific items for PAC. Future studies should include PAC related aspects.“
Comment 4: When comparing the surgical quality indicators, complication rates, etc. before and after certification, only simple statistical methods such as t-tests were used. It is suggested that the author adopt more complex statistical models, such as multivariate regression analysis, to control potential confounding factors and more accurately assess the impact of certification on the results.
Response 4: We fully acknowledge the reviewer’s concerns regarding the validity of the results without complex statistical models. We therefore added a Cox regression for survival rates: “...which did not differ significantly between the groups (x2 = 1.574, p = .8134)”. however, profound multivariate regression analysis to control for potential confounding factors could not be performed in the present study due to the limited sample size and the resulting lack of statistical power for stable multivariable tools. Therefore, only univariate statistical tests were applied. Nevertheless, we clearly recognize the importance of such analyses and plan to include multivariable modeling in future studies once a larger dataset becomes available. See note regarding literature changes in Response 2.
Comment 5: The article does not mention how to handle missing data. It should detail the methods for handling missing data and assess the potential impact of missing data on the results.
Response 5: Thank you for pointing out this important issue. Missing data in the QoL assessments were handled according to the EORTC scoring manual. Incomplete questionnaires were included if at least half of the items within a scale were answered, with missing items replaced by the mean of the completed items for that scale. Patients with completely missing QoL questionnaires were excluded from the respective time-point analyses. No additional statistical imputation was performed, as missingness was mainly due to postoperative morbidity or loss to follow-up and was considered not to be random. We updated the statistics section: “HRQoL data were pooled for 4 months (3 and 6 months) and 18 months (12 and 24 months) postoperatively and analyzed according to the EORTC-scoring manual [14]. Missing QoL data were handled according to manual, with partial questionnaires retained if ≥50% of items per scale were completed; completely missing datasets were excluded.”
Comment 6: Although the changes in quality of life scores are statistically significant, there is a lack of explanations for the clinical significance of these changes. It is suggested that the actual impact of these score changes on the quality of life of patients be discussed in light of the actual clinical situation.
Response 6: We agree with the reviewer that statistically significant differences in QoL scores do not necessarily reflect clinically relevant changes. To address this, we interpreted the results according to the established thresholds for clinical importance for the EORTC QLQ-C30. In this context, changes of 5–10 points are considered small, 10–20 points moderate, and >20 points large.
In our cohort, most score changes after certification were within the moderate range without statistical significance. Importantly, symptom control (nausea/vomiting and appetite loss) improved, suggesting better perioperative management and recovery quality. While fatigue persisted up to 18 months after surgery and reached statistical significance, this finding is also clinically relevant, as fatigue is a well-known long-term burden after major pancreatic surgery.
We acknowledge that the functioning scores were lower in the certified center cohort. Although certification led to structural improvements and standardized care pathways, this might have been accompanied by a more comprehensive documentation and higher awareness of postoperative limitations, resulting in lower reported functioning scores. From a clinical perspective, this reflects a more realistic assessment of patients’ postoperative condition rather than a true deterioration. Furthermore, despite slightly worse functioning, symptom control (nausea/vomiting and appetite loss) improved, suggesting better perioperative management and recovery quality.
Comment 7: The analysis of the results in the discussion section is rather superficial. It is necessary to delve deeply into the causes and mechanisms of changes in surgical quality, complication rate and quality of life before and after certification. Combined with relevant domestic and foreign research, the significance of these results should be further elaborated.
Response 7: We thank the reviewer for this valuable comment and have expanded the discussion to address potential causes and mechanisms underlying the observed changes in surgical quality, complication rates, and quality of life. Overall, our findings indicate that certification has led to structured quality improvements, positively affecting specific postoperative symptoms and overall care quality.
We added the following paragraphs in the discussion section:
“Certification processes are often accompanied by the implementation of standardized care pathways, structured monitoring, interdisciplinary routines, and intensified staff training — all factors known to reduce complication rates and improve symptom control.”
“Documentation effects and shifts in case mix (e.g., selective referral of more complex cases or more detailed assessment of postoperative limitations) may influence the measurement of functional scales. Especially documentation effects could explain why functioning scores appeared lower in the certified cohort.”
In the limitation section we added the following: “In addition, more elaborated documentation and the possibility that more complicated cases were treated after certification may have influenced certain functional outcomes. We therefore propose additional adjusted analyses to further quantify these mechanisms.”
Comment 8: The conclusion section holds that the certification process is valuable for improving the quality of treatment, but the basis for this conclusion seems insufficient.
Response 8: We understand the reviewer's point that a more solid basis for a generalizable conclusion would be desirable. However, we maintain that despite the considerable effort and presumed unrecovered expenses, we positively assess the certification and assume cost-effectiveness due to “expected” better overall outcomes, “which needs to be analyzed in future studies with larger cohorts”.
Comment 9: The introduction part provides a relatively brief overview of the background knowledge. It is necessary to further supplement information such as the epidemiological data of pancreatic cancer, the current situation and challenges of surgical treatment, etc., to enhance the background depth of the article.
Response 9: We appreciate the reviewer’s comment regarding the introduction. However, we believe that the current version already provides a concise yet sufficient overview of the background, including the key clinical aspects relevant to the study (e.g. still remaining high perioperative morbidity irrespective of high-volume centers). Our aim was to maintain focus and clarity while avoiding excessive detail, which is already known to the reader, such as high incidence and harsh prognosis which are beyond the scope of this manuscript. Therefore, we respectfully consider that additional epidemiological or treatment-related information, while important in general, is not essential for understanding the context of our study.

Reviewer 2 Report
Comments and Suggestions for Authors
The study investigated the impact of becoming a certified pancreatic cancer surgery center by comparing perioperative outcomes and quality of life (QoL) before and after certification. Using data from 47 patients prior to certification and 130 patients afterward, the authors found that surgical volume slightly increased, perioperative morbidity and mortality remained stable, and certain indicators such as lymph node retrieval showed non-significant improvement. QoL generally improved after surgery in both groups, though paradoxically, some functional outcomes and symptoms were less favorable post-certification, possibly due to higher patient complexity. Overall, the authors conclude that while certification did not significantly alter survival or morbidity, it strengthened multidisciplinary collaboration, quality control, and transparency of care.
Major concerns: 1) Important factors such as stage distribution, comorbidities, evolving surgical techniques, perioperative care improvements, and adjuvant treatment regimens over time are not adequately adjusted for, making it difficult to isolate the effect of certification itself. 2) The sample size is limited, particularly in the pre-certification group (n=47), which undermines statistical power and makes it difficult to detect meaningful differences. 3) In the introduction part, the authors should emphasize the dire need of diagnosis pancreatic cancer at an early stage and following references could be added: PMID: 40678708. 4) Kaplan-Meier survival analysis is presented, but the authors acknowledge no significant differences—stratification by stage, adjuvant therapy, or comorbidities could provide more nuanced insights. 5) The presentation of tables and figures is dense and difficult to follow, with overlapping data and insufficient explanation in legends. 6) The conclusions overstate the benefits of certification relative to the presented data, which show marginal or no differences in key clinical outcomes. 7) The manuscript suggests higher R1 rates post-certification reflect better pathology assessment, but this interpretation remains speculative and should be supported with standardized pathology reporting evidence. 8) Despite reporting no significant improvements in morbidity, mortality, or long-term QoL, the discussion emphasizes the value of certification. This overstates conclusions beyond what the data support. 9) Results are compared with German Cancer Society data, but without formal statistical comparison or risk adjustment, it is unclear whether this benchmarking is valid.
Author Response
Reply to referee # 2
We thank the referee for the constructive review. Here, we address all topics point by point.
Comment 1: Important factors such as stage distribution, comorbidities, evolving surgical techniques, perioperative care improvements, and adjuvant treatment regimens over time are not adequately adjusted for, making it difficult to isolate the effect of certification itself.
Response 1: We thank the reviewer for pointing this out. Since our dataset is real world data, especially with missing data accuracy in the early period, we are limited in application of statistical methods. We added therefore a paragraph in the limitation section: “This study has several limitations. First of all, the sample size. Especially the pre-certification cohort is relatively small which is caused by insufficient data availability before that, resulting in limited possibilities for applying complex statistical methods. Performing a statistically robust matching procedure would have resulted in a considerable loss of cases.”
Comment 2: The sample size is limited, particularly in the pre-certification group (n=47), which undermines statistical power and makes it difficult to detect meaningful differences.
Response 2: We totally agree with the reviewer. However, we kindly refer to Response 1.
Comment 3: In the introduction part, the authors should emphasize the dire need of diagnosis pancreatic cancer at an early stage and following references could be added: PMID: 40678708.
Response 3: We thank the reviewer for this advice. The suggested study was very interesting to read. miRNAs are of evolving importance in the early diagnosis of PAC and Serum exosomal hsa-let-7f-5p might be a promising candidate, however, since the manuscript focuses on the treatment of PAC and PAC diagnostics is beyond the scope of our manuscript, we did not cite this reference in the introduction section, however, we find it as an outlook very attractive and stated: “It is important to note that, regardless of centralized care, there is an urgent need to detect the disease early. Modern miRNA techniques can provide a possible route out of the dark [47].”.
Comment 4: Kaplan-Meier survival analysis is presented, but the authors acknowledge no significant differences—stratification by stage, adjuvant therapy, or comorbidities could provide more nuanced insights.
Response 4: We understand the reviewer's point that a more detailed statistical analysis with risk stratification could provide profound evaluation, however, multivariate regression analysis to control for potential confounding factors could not be performed in the present study due to the limited sample size and the resulting lack of statistical power for stable multivariable tools. Additionally, the data completeness before certification was limited owing to the change of the hospital information system, leading to incomplete documentation of certain baseline parameters such as adjuvant therapy. Since it is a surgical case analysis, mainly UICC stages IIA and IIB were included as stated in Table 1, “… which did not differ significantly between the groups (x2 = 1.574, p = .8134).”. We added a paragraph in the limitation section: “This study has several limitations. First of all, the sample size. Especially the pre-certification cohort is relatively small which is caused by insufficient data availability before that, resulting in limited possibilities for applying complex statistical methods. Performing a statistically robust matching procedure would have resulted in a considerable loss of cases.”
Comment 5: The presentation of tables and figures is dense and difficult to follow, with overlapping data and insufficient explanation in legends.
Response 5: This is a very important remark and we are very thankful. We adopted the Table 1 heading and the legend accordingly: “Patient characteristics, tumor histopathology, UICC stage, resection status, hospital stay and mortality within 30 days after resection for pancreatic surgery patients before A) and after certification B), stratified by year.” as well as “Values are presented as n, or mean ± SD where indicated. Histopathology: AC = adenocarcinoma, NE = neuroendocrine tumor, CA = cystadenocarcinoma, n/a = others. UICC stage: 1A–4. R-status: 0 = R0, 1 = R1, 2 = R2, x = unknown. Hospitalization: mean length of stay in days ± SD; † indicates postoperative mortality within 30 days after operation. While the mean number of PAC operations per year increased from 15 (14-18) to 19 (10-26) in the observational period, we found no significance in this deviation (p = .1859). Mean age of operated patients did not significantly differ (64.6 ± 11.0 vs. 63.7 ± 15.2; p = .4214). Patients were hospitalized for a mean of 27.4 ± 20.3 and 23.3. ± 13.6 days postoperatively, however, the median of 19 days (range 8-74 days) before, and 19 days (range 0-68) after becoming a pancreatic center did not differ significantly in the two groups, respectively (p = .3346). Total values for all years are shown.” In addition, we adopted Figure 1 legend as well.
Comment 6: The conclusions overstate the benefits of certification relative to the presented data, which show marginal or no differences in key clinical outcomes.
Response 6: We agree with the reviewer, that the conclusion could be understand rather euphemistic and with an expectation. However, since our data do not represent significant changes between the cohorts, we emphasize on procedural improvements rather than clinical outcomes. We adapted the conclusion section: “Nevertheless, certain aspects such as increased number of sufficient resected lymph nodes - although not significant - support improved patient treatment.” as well as: “Altogether, despite severe laboriousness and presumed uncovered expenses, we positively assess the certification and assume cost effectiveness due to expected better overall outcomes, which needs to be analyzed in future studies with larger cohorts.”
Comment 7: The manuscript suggests higher R1 rates post-certification reflect better pathology assessment, but this interpretation remains speculative and should be supported with standardized pathology reporting evidence.
Response 7: We thank the reviewer for this important comment. We agree that the interpretation of higher R1 rates after certification as a reflection of improved pathology assessment is, to some extent, speculative. Unfortunately, standardized pathology reporting data were not available for all cases in our cohort. However, the certification process includes mandatory adherence to structured pathology protocols and increased documentation requirements, which likely contributed to more accurate detection of microscopic residual disease such as the extinguishing of undefined Rx status due to standardized processes.
Comment 8: Despite reporting no significant improvements in morbidity, mortality, or long-term QoL, the discussion emphasizes the value of certification. This overstates conclusions beyond what the data support.
Response 8: We thank the reviewer for this observation. We acknowledge that our study did not demonstrate statistically significant improvements in overall morbidity, mortality, or long-term QoL – this might reflect decent quality even before first certification. In the discussion, we aimed to highlight the potential benefits of certification in terms of structured care pathways, standardized processes, and improved documentation, which may influence specific outcomes and perioperative management. We have revised the discussion to clarify that these points reflect potential advantages and mechanisms rather than definitive improvements in clinical endpoints, ensuring that conclusions are appropriately aligned with the observed data. In the first paragraph of the discussion section, we added: “Since we did not find significant improvements in morbidity, mortality and QoL in our study, we highlight potential benefits of certification in terms of structured care pathways, standardized processes, and improved documentation, which may influence specific outcomes and perioperative management.” Furthermore, we write: “Certification processes are often accompanied by the implementation of standardized care pathways, structured monitoring, interdisciplinary routines, and intensified staff training — all factors known to reduce complication rates and improve symptom control.”. Especially for our QoL results, we stated that “This could explain why functioning scores appeared lower in the certified cohort.” and “…undefined Rx status was extinguished from histopathological reports due to standardized processes, which could be a potential advantage rather than definitive improvements in clinical endpoints.”
Comment 9: Results are compared with German Cancer Society data, but without formal statistical comparison or risk adjustment, it is unclear whether this benchmarking is valid.
Response 9: We acknowledge that no formal statistical comparison or risk adjustment was performed with the German Cancer Society data. In the first paragraph of the discussion section, we clarify this descriptive benchmarking: “In the following, we benchmark our data with German Cancer Society data, which is intended only to provide context, without implying statistical equivalence.”
Reviewer 3 Report
Comments and Suggestions for Authors
Hospital's and/or surgeon's case-volume is crucial for better results, mainly postoperative mortality or morbidity and long term patient survival, for several complex procedures such pancreatic surgery, an association known as “The Birkmeyer effect”.
In this context centralization and certification have been proposed as means for better results in pancreatic surgery.
The authors present their experience and results after the establishment of a Certified Oncologic Center for pancreatic surgery.
Although lower mortality rates and lower surgical complications like hemorrhage, anastomotic leakage and SSI would have been expected, this was not clearly documented for all variables. Similarly there were no significant differences in the surgical performance characteristics or in all QoL parameters. It is conceivable that several factors may have influenced these results, especially when a retrospective study is concerned, but the authors should elaborate further on their result, speculate on these findings by refereeing to number of cases, number of surgeons performing resections, disease stage and so on.
There is no doubt that obtaining certification mark for pancreatic surgery results in improvement of procedures related to comprehensive and multidisciplinary diagnostic and treatment pathway. However, the lack of clear association with outcomes should be commented.
Other comments include:
Some details on their patient populations such as open/laparoscopic surgery, neoadjuvant/ adjuvant treatment should be given.
Statistical significances should be provided in the Tables and in the text to help readers to follow easily the observed differences.
Author Response
We thank the referee for the constructive review. Here, we address all topics point by point.
Comment 1: Although lower mortality rates and lower surgical complications like hemorrhage, anastomotic leakage and SSI would have been expected, this was not clearly documented for all variables.
Response 1: We thank the reviewer for this important observation. We aimed to provide an unadjusted yet transparent comparison reflecting the actual patient populations treated before and after certification. Indeed, more detailed baseline characteristics would have been beneficial, however, due to the limited sample size, especially in the pre-certification cohort (n = 47), performing a statistically robust matching procedure would have resulted in a considerable loss of cases and therefore a significant reduction in statistical power. Moreover, the data completeness before certification was limited owing to the change of the hospital information system, leading to incomplete documentation of certain baseline parameters such as pancreatic fistulas grade A to C.
Comment 2: Similarly there were no significant differences in the surgical performance characteristics or in all QoL parameters. It is conceivable that several factors may have influenced these results, especially when a retrospective study is concerned, but the authors should elaborate further on their result, speculate on these findings by refereeing to number of cases, number of surgeons performing resections, disease stage and so on.
Response 2: This is an important issue which we thank the reviewer for pointing this out. As the cause has been dealt with in response on comment 1, we hereby accentuate how we improved these remarks in the manuscript. In detail, we dealt with this concern by updating the text with the following paragraphs: “However, it must be considered that, especially in the first cohort, the data situation limited a detailed evaluation of, for example, the administration of blood transfusions or postoperative drug treatment of SSI and pancreatic fistulas.”; “The surgical procedure has been harmonized, with only two surgeons primarily per-forming the interventions.”. Moreover, we included x2 testing for consistency of UICC distribution. “Figure 1 B&C demonstrates UICC distribution in the two cohorts, which did not differ significantly between the groups (x2 = 1.574, p = .8134).” In addition, we updated the limitations section: “This study has several limitations. First of all, the sample size. Especially the pre-certification cohort is relatively small which is caused by insufficient data availability before that, resulting in limited possibilities for applying complex statistical methods. Performing a statistically robust matching procedure would have resulted in a considerable loss of cases. Therefore, data interpretability is limited by the sample size. In particular, our observations of in-hospital mortality and comparable postoperative morbidity should be viewed with caution due to the limited number of cases. Next, the QoL EORTC QLQ-C30 questionnaire mainly reflects general aspects of cancer-related QoL and lacks disease-specific items for PAC. Future studies should include PAC related aspects. In addition, more elaborated documentation and the possibility that more complicated cases were treated after certification may have influenced certain functional outcomes. We therefore propose additional adjusted analyses to further quantify these mechanisms.”
Comment 3: There is no doubt that obtaining certification mark for pancreatic surgery results in improvement of procedures related to comprehensive and multidisciplinary diagnostic and treatment pathway. However, the lack of clear association with outcomes should be commented.
Response 3: We fully agree with the reviewer that certification primarily aims to improve process quality and to ensure a structured, multidisciplinary diagnostic and therapeutic approach. However, we acknowledge that evidence directly linking certification to improved clinical outcomes remains limited. In our cohort, we observed tendencies toward enhanced perioperative management and standardization of care after certification, whereas significant differences in morbidity and mortality could not be demonstrated. This finding reflects that process optimization may precede measurable outcome improvements. We adopted the first paragraph of the discussion section: “Since we did not find significant improvements in morbidity, mortality and QoL in our study, we highlight potential benefits of certification in terms of structured care pathways, standardized processes, and improved documentation, which may influence specific outcomes and perioperative management.” Furthermore, we have addressed this limitation and the need for further studies with larger sample sizes and long-term follow-up in the revised discussion section: “Therefore, data interpretability is limited by the sample size. In particular, our observations of in-hospital mortality and comparable postoperative morbidity should be viewed with caution due to the limited number of cases.”
Comment 4: Some details on their patient populations such as open/laparoscopic surgery, neoadjuvant/ adjuvant treatment should be given.
Response 4: We agree with the reviewer, that further treatment details could clarify our results. We added: “The proportion of patients receiving neoadjuvant treatment was low in both cohorts (4% vs. 2%). Furthermore, laparoscopic procedures in our cohort were limited to distal pancreatectomies.” And in the discussion section: “The proportion of patients receiving neoadjuvant treatment was low in both cohorts reflecting the still limited role of neoadjuvant approaches during the study period. At that time, neoadjuvant therapy was not routinely established for borderline resectable PAC at our center. Current treatment strategies, however, increasingly emphasize neoadjuvant regimens for borderline and locally advanced disease, which will likely affect future perioperative and oncological outcomes.”
In addition, we added the following paragraph: “Furthermore, laparoscopic procedures in our cohort were limited to distal pancreatectomies. No minimally invasive pancreatic head resections were performed during the observation period, as these were not part of the standardized surgical portfolio at that time.”
Comment 5: Statistical significances should be provided in the Tables and in the text to help readers to follow easily the observed differences.
Response 5: Indeed, providing statistical significances will improve readers easy reading. We therefore adopted table 1 description: “While the mean number of PAC operations per year increased from 15 (14-18) to 19 (10-26) in the observational period, we found no significance in this deviation (p = .1859). Mean age of operated patients did not significantly differ (64.6 ± 11.0 vs. 63.7 ± 15.2; p = .4214). Patients were hospitalized for a mean of 27.4 ± 20.3 and 23.3. ± 13.6 days postoperatively, however, the median of 19 days (range 8-74 days) before, and 19 days (range 0-68) after becoming a pancreatic center did not differ significantly in the two groups, respectively (p = .3346). Total values for all years are shown.” In addition, we adopted Figure 1 legend as well. In accordance with response to comment 2, we added significance values for UICC distribution and survival.
Round 2
Reviewer 1 Report
Comments and Suggestions for Authors
i have no further comments.
Reviewer 2 Report
Comments and Suggestions for Authors
The authors have addressed all comments precisely and thoroughly. I have no further suggestions, and I recommend that the manuscript be accepted in its current form for publication.
Reviewer 3 Report
Comments and Suggestions for Authors
The authors grasped the essence of the comments raised by the reviewer and they responded satisfactory.
The revised manuscript has largely improved.
I have no more comments.